# Efficient Document Ranking with Learnable Late Interactions

**Ziwei Ji** [1]  **Himanshu Jain** [1]  **Andreas Veit** [1]  **Sashank J. Reddi** [1]  **Sadeep Jayasumana** [1]  **Ankit Singh Rawat** [1]
**Aditya Krishna Menon** [1]  **Felix Yu** [1]  **Sanjiv Kumar** [1]

## Abstract

Cross-Encoder (CE) and Dual-Encoder (DE) models are two fundamental approaches for predicting query-document relevance in information retrieval. To predict relevance, CE models use *joint* query-document embeddings, while DE models maintain *factorized* query-document embeddings; usually, the former has higher quality while the latter has lower latency. Recently, *late-interaction* models have been proposed to realize more favorable latency-quality trade-offs, by using a DE structure followed by a lightweight scorer based on query and document token embeddings. However, these lightweight scorers are often hand-crafted, and there is no understanding of their approximation power; further, such scorers require access to individual document token embeddings, which imposes an increased latency and storage burden over DE models. In this paper, we propose novel *learnable* late-interaction models (LITE) that resolve these issues. Theoretically, we prove that LITE is a universal approximator of continuous scoring functions, even for relatively small embedding dimension. Empirically, LITE outperforms previous late-interaction models such as ColBERT on both in-domain and zero-shot re-ranking tasks. For instance, experiments on MS MARCO passage re-ranking show that LITE not only yields a model with better generalization, but also lowers latency and requires $0.25\times$ storage compared to ColBERT.

## 1. Introduction

Transformers (Vaswani et al., 2017) have emerged as a successful model for information retrieval problems, where the goal is to retrieve and rank relevant documents for a given query (Nogueira & Cho, 2019). Two families of Transformer-based models are popular: *cross-encoder* (CE) and *dual-encoder* (DE) models. Given a (query, document) pair, CE models operate akin to a BERT-style encoder (Devlin et al., 2019): the query and document are concatenated, and sent to a Transformer encoder which outputs a relevance score (cf. Figure 1a). CE models can learn complex query-document relationships, as they allow for cross-interaction between query and document tokens.

By contrast, DE models apply two separate Transformer encoders to the query and document, respectively, producing separate query and document embedding vectors (Reimers & Gurevych, 2019). The dot product of these two vectors is used as the final relevance score (cf. Figure 1b). Compared to CE models, DE models are usually less accurate (Hofstätter et al., 2020), since the only interaction between the query and document occurs in the final dot product. However, DE models have much lower latency, since all the document embedding vectors can be pre-computed offline.

Recently, *late-interaction* models have provided alternatives with a more favorable latency-quality trade-off compared to CE and DE models. Similarly to DE models, late-interaction models also use a two-Transformer structure, but they store more information and employ additional nonlinear operations to calculate the final score. In particular, let $\mathbf{Q} \in \mathbb{R}^{P \times L_1}$ and $\mathbf{D} \in \mathbb{R}^{P \times L_2}$ denote the query and document token embeddings output by the two Transformers, i.e., there are $L_1$ query token embedding vectors and $L_2$ document token embedding vectors of dimension $P$. DE models simply pool $\mathbf{Q}$ and $\mathbf{D}$ into two vectors, and take the dot product. By contrast, ColBERT (Khattab & Zaharia, 2020) calculates the (token-wise) similarity matrix $\mathbf{Q}^\top \mathbf{D}$ and computes the final score via a sum-max reduction $\sum_i \max_j (\mathbf{Q}^\top \mathbf{D})_{i,j}$.

While the sum-max score reduction lets ColBERT achieve better accuracy than DE, it is unclear whether this hand-crafted reduction can capture arbitrary complex query-document interactions. Moreover, ColBERT can have higher latency than DE: calculating the similarity matrix $\mathbf{Q}^\top \mathbf{D}$ requires $L_1 \cdot L_2$ dot products, while the DE model only requires one dot product. Additionally, to reduce online latency, ColBERT needs to pre-compute and store the Trans-

*Equal contribution [1]Google Research, New York, USA. Correspondence to: Ziwei Ji <ziweiji@google.com>, Himanshu Jain <himj@google.com>.

Accepted to the Workshop on Advancing Neural Network Training at International Conference on Machine Learning (WANT@ICML 2024).

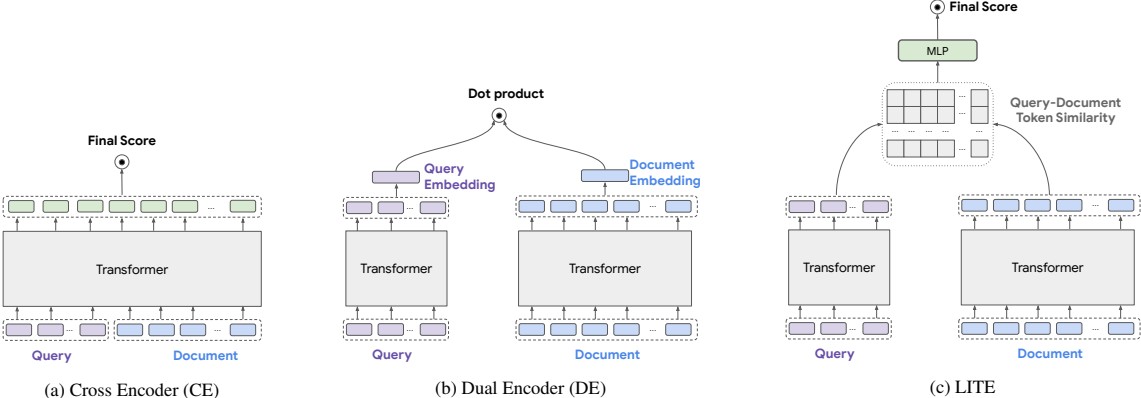

*Figure 1.* Illustration of different query-document relevance models. (a) CE models compute a joint query-document embedding by passing the concatenated query/document tokens through a single Transformer. (b) In DE models, query and document embeddings are computed separately with their respective Transformers and the relevance score is the dot product of these embeddings. (c) In the proposed LITE method, query and document token embeddings are computed similarly to DE, but instead of a dot product, we first compute the similarity matrix between each pair of query and document tokens, and pass this matrix through an MLP to produce the final relevance score.

former embedding matrix $\mathbf{D}$ for each document (Hofstätter et al., 2020; Santhanam et al., 2022). This can entail significant storage space if we decide to store a large number of document tokens, since there can be billions of documents in industry-scale information retrieval systems (Zhang & Rui, 2013; Overwijk et al., 2022). (See Section 2.3 for a detailed discussion.)

To reduce latency and storage cost, one may seek to store fewer document tokens, and/or reduce the dimension of each token embedding vector. However, it is unclear how these influence performance. In fact, such reduction can significantly hurt the accuracy of ColBERT, as we show in Section 4.4.

**Contributions.** In this work, we propose *lightweight scoring with token einsum* (LITE), which addresses the aforementioned shortcomings of existing late-interaction models. LITE applies a *lightweight and learnable non-linear transformation* on top of Transformer encoders, which corresponds to processing the (token-wise) similarity matrix $\mathbf{S} = \mathbf{Q}^\top \mathbf{D}$ via shallow multi-layer perceptron (MLP) layers (cf. Figure 1c and Section 3). In particular, we focus on a *separable LITE* scorer which applies two shared MLPs to the rows and the columns of $\mathbf{S}$ (in that order), and then projects the resulting matrix to a single scalar.

Theoretically, we rigorously establish the expressive power of LITE: we show that LITE is a universal approximator of continuous scoring functions in $\ell_2$ distance, even under tight storage constraints (cf. Theorem 3.1). To our knowledge, this is the *first formal result about the approximation power of late-interaction methods*. Further, we also construct a scoring function that cannot be approximated by a DE model with restricted embedding dimension (cf. Theorem 3.2).

Empirically, we show that LITE can systematically improve upon existing late-interaction methods like ColBERT on both in-domain benchmarks such as MS MARCO and Natural Questions (cf. Table 1), and out-of-domain benchmarks such as BEIR (cf. Table 2). Moreover, LITE can be much more accurate than ColBERT while having lower latency and storage cost (cf. Table 3).

## 2. Background

Given a query $q \in \mathcal{Q}$, the goal of information retrieval (Mitra & Craswell, 2018) is to identify the set of *relevant documents* from some corpus $\mathcal{D}$. Typically, $|\mathcal{D}|$ is large (e.g., $\mathcal{O}(10^9)$), while the number of relevant documents is small (e.g., $\mathcal{O}(10)$). A classical strategy employs a two-phase approach: in the *retrieval* phase, for moderate $K$ (e.g., $\mathcal{O}(10^3)$), one retrieves the top-$K$ documents based on a scoring function $s_{\text{ret}} \colon \mathcal{Q} \times \mathcal{D} \to \mathbb{R}$. These retrieved documents may potentially include some irrelevant documents. In the *re-ranking* phase, one applies $s_{\text{rr}} \colon \mathcal{Q} \times \mathcal{D} \to \mathbb{R}$ to re-score the $K$ documents, and keep the top scoring ones.

While $s_{\text{ret}}$ and $s_{\text{rr}}$ both score query-document relevance, they are often implemented via fundamentally different techniques. Efficiency is more important for $s_{\text{ret}}$ since we need to evaluate it over *all* documents; models such as TF-IDF and BM25 (Robertson & Zaragoza, 2009) and approximate nearest neighbor search (Guo et al., 2016b; Johnson et al., 2019; Guo et al., 2020) are used for this purpose. On the other hand, in the second phase we usually only need to re-score a few ($K \ll |\mathcal{D}|$) documents, and thus we can usually get higher accuracy by using more expensive models for $s_{\text{rr}}$. In this work, we focus on re-ranking.

## 2.1. Cross- and Dual-Encoders

*Transformers* (Vaswani et al., 2017) have been explored for both retrieval and re-ranking. Given a finite set $\mathcal{X}$, a Transformer is a function $T : \mathcal{X}^L \to \mathbb{R}^{P \times L}$, where $L$ is the sequence length and $P$ is the embedding size of each token in the sequence. A simplified Transformer network is introduced in Section 3.1 and used in our universal approximation results; for more details, we refer the readers to (Vaswani et al., 2017; Devlin et al., 2019).

To estimate query-document relevance via Transformers, one first *tokenizes* the query and document (e.g., using a SentencePiece tokeniser (Kudo & Richardson, 2018)) into $q = (q_1, \ldots, q_{L_1})$ and $d = (d_1, \ldots, d_{L_2})$. There are then two basic strategies. In *cross-encoder* (CE) models (Nogueira & Cho, 2019), we apply a single Transformer to the concatenation of $q$ and $d$, and estimate relevance with learned weights $\mathbf{w}$:

$$s(q, d) = \mathbf{w}^\top \mathsf{pool}(T(\mathsf{concat}(q, d))), \qquad (1)$$

where pool denotes a pooling strategy by which we reduce a sequence of Transformer token embeddings into a single vector. CE models can often achieve high accuracy since they can take into account interactions between the query and document tokens in *every* Transformer layer. However, they can also be expensive at inference time: we need to compute (1) for all retrieved documents, each of which involves an expensive Transformer inference (see Section 4.4 for concrete evaluations).

By contrast, in *dual-encoder* (DE) models (Karpukhin et al., 2020), we apply separate Transformers $T_1, T_2$ to the query and document, and then compute

$$s(q, d) = \mathsf{pool}(T_1(q))^\top \mathsf{pool}(T_2(d)). \qquad (2)$$

In practice, DE is usually less accurate than CE for re-ranking (Hofstätter et al., 2020), since the only interaction between the query and document is the final dot product; however, since all document embeddings $\mathsf{pool}(T_2(d))$ can be pre-computed offline, DE has much lower latency than CE.

Another idea is to apply an MLP to the concatenation of $\mathsf{pool}(T_1(q))$ and $\mathsf{pool}(T_2(d))$ (He et al., 2017). However, Rendle et al. (2020) claim that it may not be better than dot-product DE, partly because it is non-trivial to learn the dot-product operation with an MLP given the concatenated query-document embedding as the input.

## 2.2. Late-interaction scorers

Recently, there has been interest in *late-interaction* models. Similarly to DE models, such models also embed queries and documents separately into $T_1(q)$ and $T_2(d)$; however, they do not use pooling operations, but instead calculate dot products between *all pairs* of query and document token embeddings, and perform a non-linear score reduction. Formally, let us define query and document Transformer embeddings $\mathbf{Q} = (\mathbf{q}_1, \ldots, \mathbf{q}_{L_1}) := T_1(q) \in \mathbb{R}^{P \times L_1}$ and $\mathbf{D} = (\mathbf{d}_1, \ldots, \mathbf{d}_{L_2}) := T_2(d) \in \mathbb{R}^{P \times L_2}$, and let $\mathbf{S} := \mathbf{Q}^\top \mathbf{D}$ denote the similarity matrix. *ColBERT* (Khattab & Zaharia, 2020) then performs a non-linear sum-max reduction of $\mathbf{S}$:

$$s(q, d) = \sum_{i \in [L_1]} \max_{j \in [L_2]} \mathbf{q}_i^\top \mathbf{d}_j.$$

This non-linearity allows ColBERT to achieve better accuracy than DE. See (Luan et al., 2021) for a related model. Another similar approach is CEDR (MacAvaney et al., 2019), which uses multiple query-document similarity matrices (one for each layer) from pre-trained Transformers. For each query token, instead of only using the most aligned document token, Qian et al. (2022) suggest considering the top-$k$ aligned document tokens.

Instead of using similarities between all pairs of query and document token embeddings, COIL (Gao et al., 2021) only considers pairs of query and document tokens that have the same token ID, while CITADEL (Li et al., 2022) further implements a dynamic lexical routing. Li et al. (2023) use sparse token representations that can achieve competitive accuracy compared to ColBERT while being much faster. Mysore et al. (2021) suggest using co-citations as supervision for training.

Late-interaction models have precedent in the classical IR literature. For example, DRMM (Guo et al., 2016a) scores (query, document) relevance using a feedforward network on top of *count histogram* features. On top of the query-document token similarity matrix based on Word2Vec, MatchPyramid (Pang et al., 2016) applies a convolutional network, while KNRM (Xiong et al., 2017) performs kernel-based pooling. ConvKNRM (Dai et al., 2018) further uses a convolutional network on top of learned token embeddings to produce contextual embeddings. There are also relevant models from the collaborative filtering literature, such as Dziugaite & Roy (2015).

## 2.3. Limitations of existing late-interaction scorers

Late-interaction scorers such as ColBERT may be used in both the retrieval and re-ranking phases. In this paper, we focus on the latter, which has been considered in several previous works, e.g., (Hofstätter et al., 2020; Santhanam et al., 2022; Ren et al., 2021). While ColBERT can yield a more favourable latency versus quality trade-off compared to DE and CE models, there are two important limitations for its use in re-ranking.

*Limited expressivity of hand-crafted reductions.* Although prior late-interaction models include more non-linearity

compared with DE, they rely on hand-crafted score reductions, such as sum-max in ColBERT. It is unclear if these operations can capture arbitrary complex interactions among query and document tokens that define the true relevance.

*Latency and storage overhead.* Compared with CE, both DE and late-interaction models reduce latency by relying on pre-computed document (token) embeddings. For DE, this requires storing a single document embedding vector (after proper pooling, cf. (2)), and during online inference, we need to take one dot product. Unfortunately, for late-interaction models, the latency and storage cost can be much higher: suppose we use $L_1$ query embedding vectors and $L_2$ document embedding vectors to calculate the similarity matrix, then the storage cost is $L_2$ times larger than that of DE models[1], and we need to take $L_1 L_2$ dot products to obtain the similarity matrix. It is unclear how various ways to reduce the latency and storage cost affect the model performance.

In the next section, we present *LITE*, a novel late-interaction scorer that addresses both aforementioned shortcomings: (1) LITE can provably approximate a broad class of ground truth scoring functions (cf. Theorem 3.1); and (2) it is more accurate than prior late-interaction methods on both in-domain and zero-shot tasks, and is amenable to latency and storage reduction with graceful degradation in model performance (cf. Section 4).

## 3. LITE scorers

We now introduce LITE scorers. Let $\mathbf{S} := \mathbf{Q}^\top \mathbf{D} \in \mathbb{R}^{L_1 \times L_2}$ denote the similarity matrix which consists of the dot products of all query-document Transformer token embedding pairs. LITE models apply MLPs to reduce $\mathbf{S}$ to a scalar score. A natural option is to flatten $\mathbf{S}$ and then apply an MLP; we call this *flattened LITE*. On the other hand, in this paper we focus on another MLP model which we call separable LITE, motivated by separable convolution (Chollet, 2017) and MLP-Mixer (Tolstikhin et al., 2021): we first apply row-wise updates to $\mathbf{S}$, then column-wise updates, and then a linear projection to get a scalar score. Formally, we first calculate $\mathbf{S}', \mathbf{S}'' \in \mathbb{R}^{L_1 \times L_2}$ as follows: for all $1 \leq i \leq L_1$ and $1 \leq j \leq L_2$, let

$$\mathbf{S}'_{i,:} = \mathsf{LN}(\sigma(\mathbf{W}_2 \mathsf{LN}(\sigma(\mathbf{W}_1 \mathbf{S}_{i,:} + \mathbf{b}_1)) + \mathbf{b}_2)), \quad (3)$$

$$\mathbf{S}''_{:,j} = \mathsf{LN}(\sigma(\mathbf{W}_4 \mathsf{LN}(\sigma(\mathbf{W}_3 \mathbf{S}'_{:,j} + \mathbf{b}_3)) + \mathbf{b}_4)), \quad (4)$$

where $\mathsf{LN}$, $\sigma$ respectively denote layer-norm and ReLU. The final score is given by $\mathbf{w}^\top \mathsf{vec}(\mathbf{S}'')$.

---

[1]The document (token) index can be stored on disk, or in RAM. Storing in RAM significantly reduces latency, as we do not need to pay the cost of transferring embeddings from disk. Even if one were to store the index on disk, it is still of interest to reduce the total embedding size to reduce the storage and transfer cost/latency (which would scale linearly with embedding size).

Given the above definitions, it is natural to consider the expressivity of LITE. In particular, there are two fundamental questions: (1) Can we always approximate (continuous) scoring functions using LITE, even though LITE only has the similarity matrix as inputs and the original Transformer embeddings are lost? (2) Are LITE models more expressive than simpler models such as DE?

We answer these questions in the following: we show that LITE models are universal approximators of continuous scoring functions (cf. Theorem 3.1), while there exists a scoring function which cannot be approximated by a simple dot-product DE (cf. Theorem 3.2).

### 3.1. Universal approximation with LITE

We consider the Transformer architecture described by (Yun et al., 2020): it includes multiple encoding layers, each of them can be parameterized as $\mathsf{A}(\mathbf{X}) + \mathsf{FF}(\mathsf{A}(\mathbf{X}))$, where $\mathbf{X} \in \mathbb{R}^{P \times L}$ denotes the input, $\mathsf{FF}$ denotes a feedforward network, and $\mathsf{A}(\mathbf{X})$ denotes an *attention* block:

$$\mathbf{X} + \sum_{i=1}^H \mathbf{W}_o^i \mathbf{W}_v^i \mathbf{X} \mathsf{Softmax}((\mathbf{W}_k^i \mathbf{X})^\top (\mathbf{W}_q^i \mathbf{X})).$$

Here $\mathbf{W}_q^i, \mathbf{W}_k^i, \mathbf{W}_v^i \in \mathbb{R}^{C \times P}$ are *query*, *key* and *value* and projection matrices, $\mathbf{W}_o^i \in \mathbb{R}^{P \times C}$ are output projection matrices, and $H, C$ denotes the number of heads and dimension of each head. The $\mathsf{Softmax}$ function is applied to each input column.

A Transformer network defined in the above way is *permutation-equivariant* (Yun et al., 2020, Claim 1): if we permute the input token sequence, then the output token sequence is permuted in the same way. If we want the network to distinguish between different orders of tokens, we can add a positional encoding matrix $\mathbf{E} \in \mathbb{R}^{P \times L}$ to the input $\mathbf{X}$, and apply a Transformer network to $\mathbf{X} + \mathbf{E}$.

As discussed in previous sections, in the late-interaction setting, we may need to store the whole Transformer output with shape $P \times L$, which can be expensive. One solution is to apply a pooling function to reduce the number of tokens; we empirically study this method in Section 4.4, and in Theorem 3.1, we apply pooling functions to map the Transformer output in $\mathbb{R}^{P \times L}$ to $\mathbb{R}^{P \times 2}$, i.e., a sequence of two token embeddings. We show that two query tokens and two document tokens are enough for universal approximation.

Next, we define the scorers. Let $\mathcal{F}_{\sigma,n}$ denote the set of 2-layer ReLU networks with $n$-dimensional inputs and a scalar output:

$$\mathcal{F}_{\sigma,n} := \left\{ \mathbf{z} \to \mathbf{a}^\top \sigma(\mathbf{W}\mathbf{z} + \mathbf{b}) \right\},$$

where $\sigma$ denotes the ReLU activation, $\mathbf{z} \in \mathbb{R}^n$, $\mathbf{W} \in \mathbb{R}^{m \times n}$, $\mathbf{a}, \mathbf{b} \in \mathbb{R}^m$, and we allow $m$ to be arbitrarily large. We first

consider a class of flattened LITE scorers, including all two-layer ReLU networks on top of $\mathbf{S}$ that output a scalar score:

$$\mathcal{F}_{\mathrm{f}} := \{\mathbf{S} \to f(\mathsf{vec}(\mathbf{S})) | f \in \mathcal{F}_{\sigma, L_1 \cdot L_2}\}.$$

For separable LITE, we consider a simplified version of (3) and (4), but without loss of generality, as described blow: we first use a 2-layer ReLU network $f_1 : \mathbb{R}^{L_2} \to \mathbb{R}$ to reduce every row of $\mathbf{S}$ to a single scalar, and thus transform $\mathbf{S}$ into a column vector; and then we apply another 2-layer ReLU network $f_2$ to reduce this column vector into a scalar. Formally,

$$\mathcal{F}_{\mathrm{s}} := \{\mathbf{S} \to f_2(f_1(\mathbf{S})) | f_1 \in \mathcal{F}_{\sigma, L_2}, f_2 \in \mathcal{F}_{\sigma, L_1}\},$$

where we let $f_1(\mathbf{S}) \in \mathbb{R}^{L_1}$ denote the result of applying $f_1$ to every row of $\mathbf{S}$. Note that $\mathcal{F}_{\mathrm{s}}$ is a subset of the function class defined by (3) and (4) (ignoring layer normalization).

Here is our universal approximation result.

**Theorem 3.1** (Universal approximation with LITE). *Let $s : \mathbb{R}^{(P \times L_1) \times (P \times L_2)} \to \mathbb{R}$ denote a continuous scoring function with a compact support $\Omega$ and $L_1, L_2 \geq 2$. For any $\mathcal{F} \in \{\mathcal{F}_{\mathrm{f}}, \mathcal{F}_{\mathrm{s}}\}$ and any $\epsilon > 0$, there exist a scorer $f \in \mathcal{F}$, and $T_1 : \mathbb{R}^{P \times L_1} \to \mathbb{R}^{P \times 2}$ and $T_2 : \mathbb{R}^{P \times L_2} \to \mathbb{R}^{P \times 2}$, both of which consist of positional encodings, a Transformer and a pooling function, such that*

$$\int_\Omega (f(T_1(\mathbf{X})^\top T_2(\mathbf{Y})) - s(\mathbf{X}, \mathbf{Y}))^2 \mathrm{d}(\mathbf{X}, \mathbf{Y}) \leq \epsilon.$$

The proof is given in Appendix B, and is based on the "contextual mapping" techniques from (Yun et al., 2020). This result is non-trivial, since the input to LITE scorers is the similarity matrix based on only two query tokens and two document tokens; this means LITE models are universal approximators even under strong constraints on the total embedding size. In contrast, as we show in Theorem 3.2, if the total embedding size is less than $P \cdot L$, then a dot-product DE can have a large approximation error.

### 3.2. Non-universality of existing scorers

In addition to Theorem 3.1, even without positional encodings, in Theorem B.1 we show that LITE scorers are still universal approximators of arbitrary continuous scoring functions if we do not apply pooling. By contrast, without positional encodings, ColBERT can only represent permutation-equivariant ground-truth scoring functions, because the summation and maximum operations do not consider the order of input tokens. It is an open question if ColBERT is a universal approximator with positional encodings.

If we ask whether a dot-product DE can approximate arbitrary continuous functions, then we give a negative result.

**Theorem 3.2** (Limitation of DE with restricted embedding dimension). *Suppose each query and document both have $L \geq 2$ tokens. There exists a continuous ground-truth scoring function $s$ supported on $\Omega := [0, 1]^{P \times L} \times [0, 1]^{P \times L}$, such that if $O \leq PL - 1$, then for any mappings $h_1, h_2 : \mathbb{R}^{P \times L} \to \mathbb{R}^O$ that map queries and documents to $O$-dimensional vectors respectively,*

$$\int_\Omega (h_1(\mathbf{X})^\top h_2(\mathbf{Y}) - s(\mathbf{X}, \mathbf{Y}))^2 \mathrm{d}(\mathbf{X}, \mathbf{Y}) \geq \frac{1}{20}.$$

Previously Menon et al. (2022) showed that if there is no constraint on the embedding dimension, then dot-product DE is a universal approximator of continuous functions. By contrast, here we show if the DE embedding dimension is less than $PL$, there could be a constant approximation error.

## 4. Experiments

We now evaluate the proposed LITE scorer on a few standard information retrieval benchmarks, where we confirm that LITE significantly improves accuracy over existing DE and late-interaction methods on both in-domain and out-of-domain tasks. Moreover, we show that LITE remains competitive as we reduce the latency and storage cost, and in particular, LITE can achieve higher accuracy than ColBERT with less latency and $0.25 \times$ storage cost.

### 4.1. Experimental setup

**Datasets.** We evaluate scorers on both in-domain re-ranking on the MS MARCO (Nguyen et al., 2016) and Natural Questions (NQ; (Kwiatkowski et al., 2019)) datasets, and zero-shot re-ranking on the BEIR (Thakur et al., 2021) dataset.

**Training.** For training on MS MARCO, we use the official training set of triplets $(q, d_+, d_-)$, where document $d_+$ is relevant to query $q$ while $d_-$ is irrelevant. State-of-the-art methods on MS MARCO also use hard-negative mining (Qu et al., 2021; Santhanam et al., 2022); however, in this paper our focus is on comparing different late-interaction scorers, and thus we simply use the original triplet training data.

We use labels from a CE teacher model during training, as it has been observed that distillation can significantly improve performance (Santhanam et al., 2022; Menon et al., 2022). For MS MARCO, we use the scores from the T2 teacher released by Hofstätter et al. (2020). For the NQ dataset, we use a teacher model trained with 19 hard-negatives mined with BM25, following (Menon et al., 2022). For loss functions, we try the KL loss and the margin MSE loss (see Appendix A.2 for definitions of loss functions and more details of training).

**Evaluation.** For MS MARCO, we use the standard Dev set and the TREC DL 19 and 20 test sets (Craswell et al., 2020; 2021). For NQ, we utilize the version of this dataset used in (Karpukhin et al., 2020), which consists of questions, positive passages containing the correct answer, and a collection of Wikipedia passages. Re-ranking metrics are reported on the Dev query set with 200 passages containing positives, 100 BM25 hard-negatives and up to 100 random negatives, following (Menon et al., 2022). We report MRR@10 (Radev et al., 2002) and nDCG@10 (Järvelin & Kekäläinen, 2002) scores.

For BEIR, following (Thakur et al., 2021), we take the scorers trained on MS MARCO and evaluate zero-shot transfer performance. Specifically, we report evaluation results on the 14 public datasets. Thakur et al. (2021) evaluate the CE model by first retrieving 100 documents using BM25, and then calculating the nDCG@10 score for CE re-ranking; we use the same procedure.

**Models.** For the Transformer encoder, we start from a pretrained BERT model (Turc et al., 2019) which has 6 layers and 768 token dimension. For DE and late-interaction models, we let the query encoder and document encoder share weights. We use a query sequence length of 30 and a document sequence length of 200 with the Transformer. If we use all 200 document tokens to calculate the similarity matrix $\mathbf{S}$, then $\mathbf{S} \in \mathbb{R}^{30 \times 200}$. In some experiments the document sequence length is reduced in the end to save latency and storage cost; we will specify the details later. More hyperparameter details are given in Appendix A.1.

### 4.2. In-domain re-ranking on MS MARCO and NQ

In Table 1, we report MRR@10 and nDCG@10 scores for different scorers on all datasets. When calculating the similarity matrix for ColBERT and LITE, we use the original sequence length (200) and token embedding dimension (768) of the Transformer encoder. We try both the KL loss and margin MSE loss and report the better results; more details can be found in Appendix A.3.

On MS MARCO, the T2 teacher (Hofstätter et al., 2020) has Dev MRR@10 of 0.399. A DE student can only achieve MRR@10 of 0.355. Both ColBERT and separable LITE can significantly reduce this gap, but separable LITE is much better than ColBERT (0.393 vs. 0.383). We also train a 6-layer, 768-dimensional CE student using distillation from the T2 teacher; it has MRR@10 of 0.395, which is only slightly better than separable LITE. Moreover, on TREC DL 19 and 20 datasets, separable LITE also achieves better MRR@10 and nDCG@10 scores than ColBERT.

These observations generalize to the NQ dataset as well: we find that late-interaction models are much better than DE, and separable LITE is much better than ColBERT.

We also try a few ablations, including using top-$k$ aligned document tokens instead of top-1 in ColBERT, and freezing the backbone and only fine-tuning the scorers. Separable LITE achieves better accuracy than ColBERT in all cases. See Appendix A.4 for details.

### 4.3. Zero-shot re-ranking on BEIR

Table 2 presents zero-shot transfer results with ColBERT and separable LITE (from Table 1) on 14 public datasets from BEIR (Thakur et al., 2021). We also include results for the 6-layer CE model mentioned above, which is trained in the same way as other late-interaction models. We can see that separable LITE achieves better zero-shot transfer than ColBERT on 11 out of 14 datasets. CE still gives better zero-shot transfer than separable LITE, but as we show below, CE has much higher latency (cf. Table 3).

### 4.4. Results on MS MARCO with reduced latency and storage

As discussed previously, late-interaction methods may have higher latency and storage cost than DE. Suppose the Transformer encoders use $L_1$ query tokens and $L_2$ document tokens of dimension $P$, then DE only needs to take one dot product, while calculating the similarity matrix for late-interaction methods requires $L_1 L_2$ dot products. Moreover, to save online latency, we need to pre-compute and store one $P$-dimensional document embedding vector for DE, while for late-interaction methods we might need to store a $P \times L_2$ embedding matrix. This increase in storage cost is significant in industry-scale information retrieval systems, since there can be billions of documents (Zhang & Rui, 2013; Overwijk et al., 2022).

One solution is to reduce $P$ and $L_2$ to some smaller $P'$ and $L_2'$ (by projection, pooling, etc.), and then store a $P' \times L_2'$ embedding matrix for each document. Correspondingly, for each query we use $L_1$ embedding vectors of dimension $P'$, and to calculate the similarity matrix, we need $L_1 L_2'$ dot products between $P'$-dimensional vectors. This can reduce both latency and storage; below we analyze how performance drops with such reduction, and show that separable LITE remains competitive compared to ColBERT.

**Reducing the number of output document tokens.** Here, we keep the token dimension at 768 and reduce the number of output document tokens. The Transformer encoder outputs an embedding matrix $\mathbf{D} \in \mathbb{R}^{768 \times 200}$ of 200 token embeddings, and we try to reduce the number of tokens either by directly taking average of adjacent columns (average pooling), or by applying a trainable linear projection to every row of $\mathbf{D}$. We try both methods and find that separable LITE prefers learnable projection while ColBERT prefers average pooling. The results are shown in

*Table 1.* MRR@10 and nDCG@10 scores. Separable LITE achieves the best in-domain results across all benchmarks.

| Scorer | MS MARCO | | DL 2019 | | DL 2020 | | NQ | |
|---|---|---|---|---|---|---|---|---|
| | MRR | nDCG | MRR | nDCG | MRR | nDCG | MRR | nDCG |
| DE | 0.355 | 0.413 | 0.861 | 0.744 | 0.842 | 0.723 | 0.699 | 0.611 |
| ColBERT | 0.383 | 0.442 | 0.878 | 0.753 | 0.860 | 0.731 | 0.756 | 0.689 |
| Sep LITE | 0.393 | 0.452 | 0.898 | 0.765 | 0.873 | 0.756 | 0.769 | 0.693 |

*Table 2.* BEIR nDCG@10. Separable LITE is better than ColBERT on 11 out of 14 datasets.

| Dataset | ColBERT | Sep LITE | CE |
|---|---|---|---|
| T-COVID | 0.761 | 0.763 | 0.771 |
| NFCorpus | 0.356 | 0.358 | 0.361 |
| NQ | 0.525 | 0.540 | 0.552 |
| HotpotQA | 0.685 | 0.681 | 0.728 |
| FiQA-2018 | 0.330 | 0.336 | 0.346 |
| ArguAna | 0.433 | 0.424 | 0.519 |
| Touché-2020 | 0.274 | 0.305 | 0.300 |
| CQAD | 0.363 | 0.374 | 0.378 |
| Quora | 0.767 | 0.839 | 0.832 |
| DBPedia | 0.410 | 0.434 | 0.438 |
| SCIDOCS | 0.155 | 0.164 | 0.167 |
| FEVER | 0.782 | 0.788 | 0.804 |
| C-FEVER | 0.190 | 0.213 | 0.232 |
| SciFact | 0.667 | 0.633 | 0.695 |

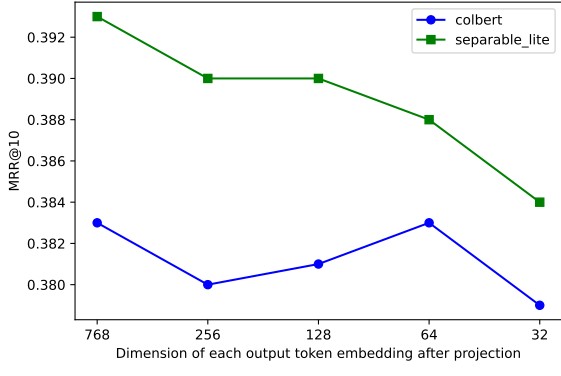

*Figure 3.* MS MARCO MRR with reduced token dimension.

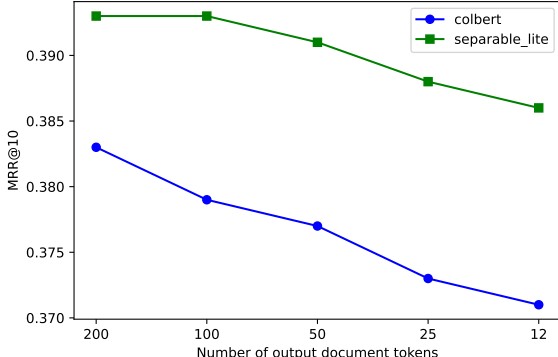

*Figure 2.* MS MARCO MRR with fewer document tokens.

Figure 2, and we can see separable LITE is more accurate than ColBERT with reduced document sequence lengths.

**Reducing token dimension.** Next, we fix the number of document tokens at 200, and reduce the dimension of each output token via learnable linear projections. The results are given in Figure 3. With different token dimension, separable LITE is always more accurate than ColBERT.

**Achieving lower latency/storage than ColBERT using LITE.** If the size of pre-computed document embedding matrix is fixed, then LITE has higher latency than ColBERT since its MLP scorer is slower than sum-max. However, since LITE is robust to embedding size reduction, it can remain more accurate than ColBERT while being more time and space efficient by using fewer document tokens. The result is shown in Table 3, together with latency of other scorers studied before.

In Table 3, we evaluate the latency of scoring relevance between 1 query and 100 documents. For CE, we use the 6-layer distilled student and evaluate the total time to calculate the joint embeddings between the query and every document. For DE, ColBERT and separable LITE, we use models from Table 1; we pre-compute the document embeddings, and evaluate the query embedding generation and scoring time. For the "small separable LITE" model, we only store 50 tokens for each document, and we also use a small MLP (we let $W_1$ in (3) have shape $(768, 50)$). In Table 3, small separable LITE only uses $0.25\times$ storage space compared with ColBERT which stores 200 document token embeddings, and it also achieves lower latency while still being much more accurate than ColBERT (0.391 vs. 0.383). We can also see that the CE latency is $100\times$ of the LITE latency, since CE cannot use offline pre-computation.

*Table 3.* Latency of different scorers.

| Scorer | Latency (in ms) | Storage | MS MARCO MRR@10 |
|---|---|---|---|
| CE (student) | 10990 | 0× | 0.395 |
| DE | 42 | 1× | 0.355 |
| ColBERT | 62 | 200× | 0.383 |
| Separable LITE | 111 | 200× | 0.393 |
| Small sep LITE | 56 | 50× | 0.391 |

### 4.5. Comparison with KNRM

KNRM (Xiong et al., 2017) is one popular pre-Transformer scorer; it calculates the similarity matrix using Word2Vec embeddings, and then apply kernel pooling. It has been applied to MS MARCO in a few recent works (Khattab & Zaharia, 2020; Hofstätter et al., 2020); however, KNRM only achieves low accuracy, likely because the underlying encoders are non-pretrained shallow Transformers. In this work, we try to apply KNRM with the same pretrained BERT encoder as other scorers. We find that KNRM can achieve similar accuracy to ColBERT overall, but separable LITE is still better than KNRM over all benchmarks; see Appendix A.5 for details.

## 5. Conclusion

In this work, we propose LITE models that can provably approximate any continuous scoring functions. We also show that LITE outperforms prior late-interaction models in both in-domain and zero-shot reranking. Moreover, LITE is still more accurate than prior methods if we try to reduce the storage cost of pre-computed document embeddings, and in particular LITE can achieve higher accuracy with less latency and storage cost.

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

# A. Experimental details

## A.1. Hyper-parameters

The main hyperparameters for LITE are the MLP widths. For Separable LITE (cf. (3) and (4)), if the input dot-product matrix has shape $L_1 \times L_2$, then $\mathbf{W}_1$ has shape $(m_2, L_2)$, $\mathbf{W}_2$ has shape $(L_2, m_2)$, $\mathbf{W}_3$ has shape $(m_1, L_1)$, and $\mathbf{W}_4$ has shape $(L_1, m_1)$. In this work, we let $m_1 = 360$ and $m_2 = 2400$ in most experiments for simplicity, but we also note that much smaller widths can already give a high accuracy while also reducing the latency (cf. Table 3).

## A.2. Training details

Here we first define the loss functions used in our experiments.

For simplicity, let us first consider the triplet setting, where we are given a query $q$, a positive document $d_+$, and a negative document $d_-$. Suppose the teacher score is given by $\mathbf{t} = (t_+, t_-)$, and the student score is $\mathbf{s} = (s_+, s_-)$. The margin MSE loss is defined as $((t_+ - t_-) - (s_+ - s_-))^2$, i.e., it calculates the teacher score margin and student score margin, and applies a squared loss. The KL loss first calculates the teacher and student probability distributions as below

$$\mathbf{p}^{(t)} = \left( \frac{\exp(t_+)}{\exp(t_+) + \exp(t_-)}, \frac{\exp(t_-)}{\exp(t_+) + \exp(t_-)} \right),$$
$$\mathbf{p}^{(s)} = \left( \frac{\exp(s_+)}{\exp(s_+) + \exp(s_-)}, \frac{\exp(s_-)}{\exp(s_+) + \exp(s_-)} \right),$$

and then calculates the KL divergence $\mathrm{KL}(\mathbf{p}^{(t)} || \mathbf{p}^{(s)})$.

In our NQ experiments, we use one positive document and multiple negative documents. In this case the KL loss is defined similarly, while for the margin MSE loss we consider the margins between the positive document and every negative document. Formally, suppose there are $N$ documents, the first one is positive while the remaining ones are negative, and let $t_i$ and $s_i$ denote the teacher and student scores for the $i$-th document, then we consider

$$\sum_{i=2}^{N} ((t_1 - t_i) - (s_1 - s_i))^2.$$

It is also an interesting open direction to try other training frameworks, such as sRank (Zhu et al., 2023).

On the optimization algorithm, we use AdamW (Loshchilov & Hutter, 2019) with batch size 128, peak learning rate $2.8 \times 10^{-5}$, weight decay 0.01, and 1.5 million steps. We use a linear learning rate warm up of 30000 steps, then a linear learning rate decay.

## A.3. Results with different loss functions

Here we present results on different scorers and loss functions.

First, Table 4 includes results on MS MARCO.

Table 4. MS MARCO Dev MRR@10. Separable LITE achieves the best results among factorized (non-CE) models.

| Scorer | KL | Margin MSE |
|---|---|---|
| CE student | 0.394 | 0.395 |
| DE | 0.355 | 0.350 |
| ColBERT | 0.383 | 0.378 |
| Separable LITE | 0.388 | 0.393 |

For context, the T2 teacher (Hofstätter et al., 2020) achieves a Dev MRR@10 of 0.399. Even a CE student (with 6 layers and token dimension 768) cannot match this teacher performance: the best MRR@10 we get is 0.395.

We also note that separable LITE get good results for both the KL loss and margin MSE loss, while other scorers seem to prefer only one loss. It is interesting to understand the effects of loss functions.

*Table 5.* Natural Questions Dev MRR@10. Separable LITE achieves the best results both in direct training and distillation settings.

| Scorer | Cross Entropy (one-hot labels) | KL (distillation) | Margin MSE |
|---|---|---|---|
| DE | 0.678 | 0.699 | 0.699 |
| ColBERT | 0.690 | 0.754 | 0.756 |
| Separable LITE | 0.710 | 0.741 | 0.769 |

Table 5 includes results on NQ. Here we report results in two settings: direct training with 1-hot labels and the cross entropy loss, and distillation training with the KL loss and margin MSE loss. Separable LITE achieves the best results for both the cross-entropy loss and margin MSE loss; although ColBERT performs better with the KL loss, it gives lower scores than the margin MSE loss.

### A.4. Model ablations

**Using top-$k$ aligned document tokens in ColBERT.** Given query Transformer embedding vectors $\mathbf{q}_1, \ldots, \mathbf{q}_{L_1}$ and document Transformer embedding vectors $\mathbf{d}_1, \ldots, \mathbf{d}_{L_2}$, recall that ColBERT performs a sum-max reduction:

$$\sum_{i \in [L_1]} \max_{j \in [L_2]} \mathbf{q}_i^\top \mathbf{d}_j.$$

In other words, for each query token $\mathbf{q}_i$, ColBERT finds the most-aligned document embedding vector and includes their dot-product in the score. Qian et al. (2022) suggest using top-$k$ aligned document tokens for each query token; here we try $k = 2, 4, 8$ on MS MARCO, but do not notice significant improvement compared with $k = 1$.

| $k$ | 1 | 2 | 4 | 8 |
|---|---|---|---|---|
| MRR@10 | 0.383 | 0.378 | 0.380 | 0.382 |

*Table 6.* Dev MRR@10 on MS MARCO with different values of $k$. We find that $k = 1$ (i.e., the original ColBERT) is better than other options we try ($k = 2, 4, 8$).

**Freezing query and document encoders.** Recall that we use pretrained BERT models for query and document encoding, and moreover in all experiments above we also fine-tune the pretrained Transformers on MS MARCO and NQ. Here we explore performance of different scorers when the query and document Transformer encoders are frozen (i.e., pre-trained but not fine-tuned on MS MARCO).

When the query and document encoders are frozen, ColBERT does not require any additional fine-tuning since the sum-max function does not include any weights. In this case, ColBERT can achieve Dev MRR@10 score 0.112 on MS MARCO.

For separable LITE, if we freeze the query and document Transformer encoders and only fine tune the separable LITE scorer (i.e., $\mathbf{W}_1, \mathbf{b}_1, \mathbf{W}_2, \mathbf{b}_2, \mathbf{W}_3, \mathbf{b}_3, \mathbf{W}_4, \mathbf{b}_4$ in (3) and (4)), then it can achieve Dev MRR@10 score 0.188 on MS MARCO, which is much better than ColBERT.

### A.5. KNRM results

For KNRM, following (Xiong et al., 2017), we use $K = 11$ kernels, where $\mu_1 = 0.9$, $\mu_2 = 0.7$, ..., $\mu_{10} = -0.9$ with $\sigma_1 = \cdots = \sigma_{10} = 0.1$, and $\mu_{11} = 1.0$ with $\sigma_{11} = 10^{-3}$. We hold $\mu_k$ and $\sigma_k$ fixed and only train $\mathbf{w}$.

We report MRR@10 and nDCG@10 scores on in-domain tasks in Table 7. KNRM achieves similar scores to ColBERT overall, while separable LITE is more accurate than KNRM on all benchmarks.

Moreover, separable LITE is much better than KNRM on zero-shot transfer: it is better than KNRM on 12 out of 14 datasets, as shown in Table 8.

*Table 7.* MRR@10 and nDCG@10 scores for in-domain tasks. KNRM is similar to ColBERT overall, while worse than separable LITE on all tasks.

| Scorer | MS MARCO | | DL 2019 | | DL 2020 | | NQ | |
|---|---|---|---|---|---|---|---|---|
| | MRR | nDCG | MRR | nDCG | MRR | nDCG | MRR | nDCG |
| ColBERT | 0.383 | 0.442 | 0.878 | 0.753 | 0.860 | 0.731 | 0.756 | 0.689 |
| KNRM | 0.390 | 0.448 | 0.859 | 0.744 | 0.858 | 0.730 | 0.759 | 0.682 |
| Sep LITE | 0.393 | 0.452 | 0.898 | 0.765 | 0.873 | 0.756 | 0.769 | 0.693 |

*Table 8.* BEIR nDCG@10. Separable LITE is better than KNRM on 12 out of 14 datasets.

| Dataset | KNRM | Separable LITE |
|---|---|---|
| T-COVID | 0.741 | 0.763 |
| NFCorpus | 0.353 | 0.358 |
| NQ | 0.526 | 0.540 |
| HotpotQA | 0.678 | 0.681 |
| FiQA-2018 | 0.328 | 0.336 |
| ArguAna | 0.446 | 0.424 |
| Touché-2020 | 0.301 | 0.305 |
| CQAD | 0.367 | 0.374 |
| Quora | 0.239 | 0.839 |
| DBPedia | 0.420 | 0.434 |
| SCIDOCS | 0.159 | 0.164 |
| FEVER | 0.715 | 0.788 |
| C-FEVER | 0.199 | 0.213 |
| SciFact | 0.645 | 0.633 |

# B. Proof of Theorem 3.1

Here we prove Theorem 3.1. We first restate it here and also include a universal approximation result without positional encodings.

**Theorem B.1** (Universal approximation with LITE). *Let $s : \mathbb{R}^{(P \times L_1) \times (P \times L_2)} \to \mathbb{R}$ denote a continuous scoring function with a compact support $\Omega$ and $L_1, L_2 \geq 2$. For any $\mathcal{F} \in \{\mathcal{F}_f, \mathcal{F}_s\}$ and any $\epsilon > 0$, there exists a query Transformer $T_1 : \mathbb{R}^{P \times L_1} \to \mathbb{R}^{P \times L_1}$, a document Transformer $T_2 : \mathbb{R}^{P \times L_2} \to \mathbb{R}^{P \times L_2}$, and a scorer $f \in \mathcal{F}$, such that*

$$\int_\Omega \left( f\left( T_1(\mathbf{X})^\top T_2(\mathbf{Y}) \right) - s(\mathbf{X}, \mathbf{Y}) \right)^2 \mathrm{d}(\mathbf{X}, \mathbf{Y}) \leq \epsilon.$$

*Under the same conditions, there also exist positional encoding matrices $\mathbf{E} \in \mathbb{R}^{P \times L_1}$ and $\mathbf{F} \in \mathbb{R}^{P \times L_2}$, a query Transformer $T_1 : \mathbb{R}^{P \times L_1} \to \mathbb{R}^{P \times L_1}$ and a pooling function $\mathsf{pool}_1 : \mathbb{R}^{P \times L_1} \to \mathbb{R}^{P \times 2}$, a document Transformer $T_2 : \mathbb{R}^{P \times L_2} \to \mathbb{R}^{P \times L_2}$ and a pooling function $\mathsf{pool}_2 : \mathbb{R}^{P \times L_2} \to \mathbb{R}^{P \times 2}$, and a scorer $f \in \mathcal{F}$, such that*

$$\int_\Omega \left( f\left( \mathsf{pool}_1(T_1(\mathbf{X} + \mathbf{E}))^\top \mathsf{pool}_2(T_2(\mathbf{Y} + \mathbf{F})) \right) - s(\mathbf{X}, \mathbf{Y}) \right)^2 \mathrm{d}(\mathbf{X}, \mathbf{Y}) \leq \epsilon.$$

Our proof is based on the analysis of (Yun et al., 2020): they showed that Transformer networks are universal approximators of continuous and compactly-supported sequence-to-sequence functions. In our case, we need to show universal approximation with the dot-product matrix; to this end, we actually need a few technical lemmas from (Yun et al., 2020), as detailed below.

Without loss of generality, we assume the support of the ground-truth scoring function is contained in $[0, 1)^{P \times L_1} \times [0, 1)^{P \times L_2}$. The first step is to replace the ground-truth scoring function $s$ with a piece-wise constant function: let $\delta > 0$ be small enough, and let

$$s_\delta(\mathbf{X}, \mathbf{Y}) := \sum_{\mathbf{X}' \in \mathbb{G}_\delta, \mathbf{Y}' \in \mathbb{H}_\delta} s(\mathbf{X}', \mathbf{Y}') \mathbb{1}\left[ \mathbf{X} \in \mathbb{C}_{\mathbf{X}'} \text{ and } \mathbf{Y} \in \mathbb{C}_{\mathbf{Y}'} \right], \tag{5}$$

where $\mathbf{X} \in [0,1)^{P \times L_1}$, and $\mathbf{Y} \in [0,1)^{P \times L_2}$, and $\mathbb{G}_\delta := \{0, \delta, \ldots, 1 - \delta\}^{P \times L_1}$, and $\mathbb{H}_\delta := \{0, \delta, \ldots, 1 - \delta\}^{P \times L_2}$, and $\mathbb{C}_{\mathbf{X}'} := \prod_{j=1}^{P} \prod_{k=1}^{L_1} [X'_{j,k}, X'_{j,k} + \delta)$, and $\mathbb{C}_{\mathbf{Y}'} := \prod_{j=1}^{P} \prod_{k=1}^{L_2} [Y'_{j,k}, Y'_{j,k} + \delta)$. Since $s$ is continuous, if $\delta$ is small enough, it holds that $s_\delta$ is a good approximation of $s$.

Next we follow (Yun et al., 2020) and try to approximate $s_\delta$ using LITE models based on *modified* Transformers. Recall that a standard Transformer uses softmax in attention layers and ReLU activation in MLPs; by contrast, in a modified Transformer, we use hardmax in attention layers, and in MLPs we are allowed to use activation functions from $\Phi$ which consists of piece-wise linear functions with at most three pieces where at least one piece is a constant. Such a modified Transformer can then be approximated by a standard Transformer (Yun et al., 2020, Lemma 9).

Here are two key lemmas from (Yun et al., 2020). For simplicity, we state them for the query Transformer, but they will also be applied to the document Transformer.

The following lemma ensures that there exists a modified Transformer that can quantize the input domain, and thus we can just work with $\mathbb{G}_\delta$. Similarly, on the document side, we can focus on $\mathbb{H}_\delta$.

**Lemma B.2** ((Yun et al., 2020) Lemma 5). *There exists a feedforward network $g_q : [0,1)^{P \times L_1} \to \mathbb{G}_\delta$ with activations from $\Phi$, such that for any entry $1 \le i \le P$ and any $1 \le j \le L_1$, it holds that $g_q(\mathbf{X})_{i,j} = k\delta$ if $X_{i,j} \in [k\delta, (k+1)\delta)$, $k = 0, \ldots, 1/\delta - 1$.*

The following lemma ensures the existence of a modified Transformer that can implement a "contextual mapping": roughly speaking, it means each token of the Transformer output is a a unique Hash encoding of the whole input token sequence. Below is a formal statement.

**Lemma B.3** ((Yun et al., 2020) Lemma 6). *Consider the following subset of $\mathbb{G}_\delta$:*

$$\widetilde{\mathbb{G}}_\delta := \{\mathbf{X} \in \mathbb{G}_\delta | \mathbf{X}_{:,i} \neq \mathbf{X}_{:,j} \text{ for all } i \neq j\}.$$

*If $L_1 \ge 2$ and $\delta \le 1/2$, then there exists an attention network $g_c : \mathbb{R}^{P \times L_1} \to \mathbb{R}^{P \times L_1}$ with the hardmax operator, a vector $\mathbf{u} \in \mathbb{R}^P$, constants $t_l, t_r$ with $0 < t_l < t_r$, such that $\alpha(\mathbf{X}) := \mathbf{u}^\top g_c(\mathbf{X})$ satisfies the following conditions:*

1. *For any $\mathbf{X} \in \widetilde{\mathbb{G}}_\delta$, all entries of $\alpha(\mathbf{X})$ are different.*

2. *For any $\mathbf{X}, \mathbf{X}' \in \widetilde{\mathbb{G}}_\delta$ such that $\mathbf{X}'$ is not a permutation of $\mathbf{X}$, all entries of $\alpha(\mathbf{X})$, $\alpha(\mathbf{X}')$ are different.*

3. *For any $\mathbf{X} \in \widetilde{\mathbb{G}}_\delta$, all entries of $\alpha(\mathbf{X})$ are in $[t_l, t_r]$.*

4. *For any $\mathbf{X} \in \mathbb{G}_\delta \setminus \widetilde{\mathbb{G}}_\delta$, all entries of $\alpha(\mathbf{X})$ are outside $[t_l, t_r]$.*

For the document side, consider

$$\widetilde{\mathbb{H}}_\delta := \{\mathbf{X} \in \mathbb{H}_\delta | \mathbf{Y}_{:,i} \neq \mathbf{Y}_{:,j} \text{ for all } i \neq j\}.$$

Lemma B.3 also ensures the existence of an attention network $h_c : \mathbb{R}^{P \times L_2} \to \mathbb{R}^{P \times L_2}$ with the hardmax operator, a vector $\mathbf{v} \in \mathbb{R}^P$, constants $s_l, s_r$ with $0 < s_l < s_r$, such that $\beta(\mathbf{Y}) := \mathbf{v}^\top h_c(\mathbf{Y})$ satisfies similar conditions. Also note that for small enough $\delta$, we can neglect $\mathbb{G}_\delta \setminus \widetilde{\mathbb{G}}_\delta$ and $\mathbb{H}_\delta \setminus \widetilde{\mathbb{H}}_\delta$, since $|\mathbb{G}_\delta \setminus \widetilde{\mathbb{G}}_\delta| = O\left(\delta^P |\mathbb{G}_\delta|\right)$ and $|\mathbb{H}_\delta \setminus \widetilde{\mathbb{H}}_\delta| = O\left(\delta^P |\mathbb{H}_\delta|\right)$.

Now we are ready to prove Theorem B.1. We first consider the case without positional encodings.

**Analysis without positional encodings.** Note that for $\mathbf{X} \in \widetilde{\mathbb{G}}_\delta$ and $\mathbf{Y} \in \widetilde{\mathbb{H}}_\delta$, it holds that $\alpha(\mathbf{X})$ and $\beta(\mathbf{Y})$ already include enough information to determine the score. However, in LITE models, the final score is calculated only based on dot products between query embedding vectors and document embedding vectors. As a result, we need to first insert $\mathbf{u}$ and $\mathbf{v}$ into the Transformer embeddings. The following lemma handles this issue: there exists a feedforward network such that for each $\mathbf{X} \in \widetilde{\mathbb{G}}_\delta$, it replaces one token in $g_c(\mathbf{X})$ with $\mathbf{v}$ while keeping other tokens unchanged.

**Lemma B.4.** *Consider the activation function $\varphi$ with $\varphi(z) = 1$ if $0 \le z \le 1$, and $\varphi(z) = 0$ if $z < 0$ or $z > 1$. There exists a feedforward network $g_v : \mathbb{R}^P \to \mathbb{R}^P$ with activation $\varphi$ such that for any $\mathbf{X} \in \widetilde{\mathbb{G}}_\delta$, let $i := \arg\min_j \alpha(\mathbf{X})_j$, then $g_v(g_c(\mathbf{X})_{:,i}) = \mathbf{v}$, while for $j \neq i$, it holds that $g_v(g_c(\mathbf{X})_{:,j}) = g_c(\mathbf{X})_{:,j}$.*

*Proof.* For any $\mathbf{X} \in \widetilde{\mathbb{G}}_\delta$ and any $i$, $1 \leq i \leq L_1$, Lemma B.3 ensures that there exists constants $l(\mathbf{X}, i)$ and $r(\mathbf{X}, i)$ such that $0 < l(\mathbf{X}, i) < \alpha(\mathbf{X})_i < r(\mathbf{X}, i)$, and that $[l(\mathbf{X}, i), r(\mathbf{X}, i)]$ does not contain other entries in $\alpha(\mathbf{X})$, and moreover $[l(\mathbf{X}, i), r(\mathbf{X}, i)]$ does not contain entries from $\alpha(\mathbf{X}')$ for $\mathbf{X}' \in \widetilde{\mathbb{G}}_\delta$ which is not a permutation of $\mathbf{X}$. For this $(\mathbf{X}, i)$ pair, if $i := \arg\min_j \alpha(\mathbf{X})_j$, we construct the following neuron

$$\psi_{\mathbf{X},i}(\mathbf{z}) := \varphi\left(\frac{1}{r(\mathbf{X}, i) - l(\mathbf{X}, i)}\left(\mathbf{u}^\top \mathbf{z} - l(\mathbf{X}, i)\right)\right)\mathbf{v},$$

otherwise let

$$\psi_{\mathbf{X},i}(\mathbf{z}) := \varphi\left(\frac{1}{r(\mathbf{X}, i) - l(\mathbf{X}, i)}\left(\mathbf{u}^\top \mathbf{z} - l(\mathbf{X}, i)\right)\right)g_c(\mathbf{X})_{:,i}.$$

The full network is the sum of all such neurons

$$g_v(\mathbf{z}) := \sum_{\mathbf{X} \in \widetilde{\mathbb{G}}_\delta, 1 \leq i \leq L_1} \psi_{\mathbf{X},i}(\mathbf{z}),$$

which satisfies the requirement of Lemma B.4. $\qquad\square$

Lemma B.4 is stated for the query side; on the document side, it also follows that there exists a feedforward network $h_u$ that can replace one token in the embeddings given by $h_c$ by $\mathbf{u}$. Then we are ready to prove Theorem B.1 without positional encodings.

*Proof of Theorem B.1, no positional encodings.* In this proof, we will focus on $\mathbf{X} \in \widetilde{\mathbb{G}}_\delta$ and $\mathbf{Y} \in \widetilde{\mathbb{H}}_\delta$ as ensured by Lemmas B.2 and B.3. We also use notation introduced in Lemmas B.3 and B.4.

First consider $\mathbf{u}$ and $\mathbf{v}$ given by Lemma B.3. Without loss of generality, we can assume $\mathbf{u}^\top \mathbf{v} \leq 0$; if $\mathbf{u}^\top \mathbf{v} > 0$, we will replace $\mathbf{v}$ with $-\mathbf{v}$ and replace $h_c(\mathbf{Y})$ with $-h_c(\mathbf{Y})$, which ensures $\mathbf{u}^\top \mathbf{v} \leq 0$, and moreover the conclusions of Lemma B.3 still hold. In detail, in the construction of $g_v$, we use $-\mathbf{v}$ instead of $\mathbf{v}$, while in the construction of $h_u$, we use $-h_c(\mathbf{Y})$ instead of $h_c(\mathbf{Y})$. As a result, in the following we assume $\mathbf{u}^\top \mathbf{v} \leq 0$.

Recall that for $\mathbf{X} \in \widetilde{\mathbb{G}}_\delta$, the range of $\mathbf{u}^\top g_c(\mathbf{X})$ is denoted by $[t_l, t_r]$ with $0 < t_l < t_r$, while for $\mathbf{Y} \in \widetilde{\mathbb{H}}_\delta$, the range of $\mathbf{v}^\top h_c(\mathbf{Y})$ is denoted by $[s_l, s_r]$ with $0 < s_l < s_r$. Define

$$M := \max_{\mathbf{X} \in \widetilde{\mathbb{G}}_\delta} \max_{\mathbf{Y} \in \widetilde{\mathbb{H}}_\delta} \max_{i,j} \left|g_c(\mathbf{X})_{:,i}^\top h_c(\mathbf{Y})_{:,j}\right|.$$

In the following, we will assume $t_l > M$ and $s_l > t_r$ without loss of generality; if these conditions do not hold, we can let $\lambda_1, \lambda_2$ be large enough such that $\lambda_1 t_l > M$ and $\lambda_2 s_l > \lambda_1 t_r$, and scale $\mathbf{u}$ to $\lambda_1 \mathbf{u}$, and scale $\mathbf{v}$ to $\lambda_2 \mathbf{v}$.

Given $\mathbf{X} \in \widetilde{\mathbb{G}}_\delta$ and $\mathbf{Y} \in \widetilde{\mathbb{H}}_\delta$, we consider $\mathbf{Q} = g_v(g_c(\mathbf{X})) \in \mathbb{R}^{P \times L_1}$, and $\mathbf{D} = h_u(h_c(\mathbf{Y})) \in \mathbb{R}^{P \times L_2}$, and the dot-product matrix $\mathbf{S} := \mathbf{Q}^\top \mathbf{D} \in \mathbb{R}^{L_1 \times L_2}$. Lemma B.4 ensures that $\mathbf{Q}$ has one column equal to $\mathbf{v}$, while $\mathbf{D}$ has one column equal to $\mathbf{u}$.

Let $\mathbf{q}$ denote an arbitrary column of $\mathbf{Q}$ other than $\mathbf{v}$, and let $\mathbf{d}$ denote an arbitrary column of $\mathbf{D}$ other than $\mathbf{u}$. Due to previous discussion, we have $\mathbf{v}^\top \mathbf{d} \geq s_l > t_r \geq \mathbf{q}^\top \mathbf{u}$, and therefore we can distinguish them. Additionally $\mathbf{q}^\top \mathbf{u} \geq t_l > M$, and thus we can distinguish it from other entries of $\mathbf{S}$, including $\mathbf{v}^\top \mathbf{u} \leq 0$.

Now let us examine $\mathbf{S}$ in detail. Suppose $\mathbf{Q}_{:,i} = \mathbf{v}$ and $\mathbf{D}_{:,j} = \mathbf{u}$ for some $1 \leq i \leq L_1$ and $1 \leq j \leq L_2$. Then

$$\mathbf{S}_{i,:} = (\mathbf{Q}^\top \mathbf{D})_{i,:} = [\mathbf{v}^\top \mathbf{d}_1, \cdots, \mathbf{v}^\top \mathbf{u}, \cdots, \mathbf{v}^\top \mathbf{d}_{L_2}],$$

and

$$\mathbf{S}_{:,j} = [\mathbf{q}_1^\top \mathbf{u}, \cdots, \mathbf{v}^\top \mathbf{u}, \cdots, \mathbf{q}_{L_1}^\top \mathbf{u}]^\top.$$

The previous scaling allows us to find $\mathbf{S}_{i,:}$ and $\mathbf{S}_{:,j}$. Lemma B.3 ensures that every element of $\mathbf{S}_{i,:}$ other than $\mathbf{v}^\top \mathbf{u}$ can uniquely determine the set of columns of the document input $\mathbf{Y}$, but not the order of columns since Transformers without positional encodings are permutation-equivariant (Yun et al., 2020, Claim 1). However, all elements of $\mathbf{S}_{i,:}$ together are able

to determine the exact order of columns of $\mathbf{Y}$. Similarly, $\mathbf{S}_{:,j}$ as a whole can determine the exact query input $\mathbf{X}$, including the order of columns. Consequently, $\mathbf{S}$ can uniquely determine the input pair $(\mathbf{X}, \mathbf{Y})$, and also the ground-truth score $s_\delta(\mathbf{X}, \mathbf{Y})$.

For flattened LITE, note that $\widetilde{\mathbb{G}}_\delta$ and $\widetilde{\mathbb{H}}_\delta$ are both finite, and thus the set of possible dot-product matrix

$$\left\{\mathbf{Q}^\top \mathbf{D} \middle| \mathbf{Q} = g_{\mathrm{v}}(g_{\mathrm{c}}(\mathbf{X})), \mathbf{D} = h_{\mathrm{u}}(h_{\mathrm{c}}(\mathbf{Y})), \mathbf{X} \in \widetilde{\mathbb{G}}_\delta, \mathbf{Y} \in \widetilde{\mathbb{H}}_\delta\right\}$$

is also finite. Moreover, each dot-product matrix uniquely determines the ground-truth score, as discussed above. Therefore there exists a 2-layer ReLU network that uniformly approximates an interpolations of these scores (Cybenko, 1989; Funahashi, 1989; Hornik et al., 1989), which finishes the proof.

For separable LITE, recall that we first apply a shared MLP $f_1$ to reduce every row of $\mathbf{S}$ to a scalar, and thus get a column vector; then we apply another MLP $f_2$ to reduce this column vector to a final score. Now let $\psi$ denote an injection from $\widetilde{\mathbb{H}}_\delta$ to $[t_r + 1, t_r + 2]$, i.e., for any $\mathbf{Y}, \mathbf{Y}' \in \widetilde{\mathbb{H}}_\delta$, we have $\psi(\mathbf{Y}), \psi(\mathbf{Y}') \in [t_r + 1, t_r + 2]$, and $\psi(\mathbf{Y}) \neq \psi(\mathbf{Y}')$. There exists such a $\psi$ since $\widetilde{\mathbb{H}}_\delta$ is finite.

Now if the $i$-th column of $\mathbf{Q}$ is $\mathbf{v}$, then we let $f_1$ map $\mathbf{S}_{i,:}$ to $\psi(\mathbf{Y})$; this is well-defined since $\mathbf{S}_{i,:}$ uniquely determines $\mathbf{Y}$, as discussed above. For any $i' \neq i$, we let $f_1$ map $\mathbf{S}_{i',:}$ to $\mathbf{q}_{i'}^\top \mathbf{u} \in [t_l, t_r]$. Note that by our construction, $f_1(\mathbf{S}_{i,:}) \geq t_r + 1 > t_r \geq f_1(\mathbf{S}_{i',:})$. As a result, $f_1(\mathbf{S})$ can uniquely determines $(\mathbf{X}, \mathbf{Y})$, and thus there exists another MLP $f_2$ which can approximate the ground-truth score $s_\delta$. $\qquad\square$

**Analysis with positional encodings.** Here we consider the case with positional encodings. Following (Yun et al., 2020), we will use fixed positional encodings: let $\mathbf{1}$ denote the $P$-dimensional all-ones vector, and let $\mathbf{E} \in \mathbb{R}^{P \times L_1}$ denote the matrix whose $j$-th column is given by $(j - 1)\mathbf{1}$, and similarly let $\mathbf{F} \in \mathbb{R}^{P \times L_2}$ denote the matrix whose $j$-th column is given by $(j - 1)\mathbf{1}$. Given input $\mathbf{X} \in [0, 1)^{P \times L_1}$ and $\mathbf{Y} \in [0, 1)^{P \times L_2}$, we transform them to $(\mathbf{X} + \mathbf{E})/L_1$ and $(\mathbf{Y} + \mathbf{F})/L_2$. Note that after the transformation, it holds that $(\mathbf{X} + \mathbf{E})/L_1 \in \prod_{i=1}^{P}\prod_{j=1}^{L_1}[(j-1)/L_1, j/L_1)$; in other words, different columns of $(\mathbf{X} + \mathbf{E})/L_1$ have different ranges.

We can now invoke our earlier analysis. Let $\delta = 1/(nL_1L_2)$ for some large enough integer $n$ such that the approximation error in (5) is small enough. Then Lemma B.2 implies there exist feedforward networks $g_{\mathrm{q}}$ and $h_{\mathrm{q}}$ that can quantize the input domains to $\mathbb{G}_\delta = \{0, \delta, \cdots, 1 - \delta\}^{P \times L_1}$ and $\mathbb{H}_\delta = \{0, \delta, \cdots, 1 - \delta\}^{P \times L_2}$. Combined with the positional encodings, we only need to consider the following input domains:

$$\mathbb{G}_{\delta,\mathrm{pe}} := \left\{g_{\mathrm{q}}((\mathbf{X} + \mathbf{E})/L_1) \middle| \mathbf{X} \in [0, 1)^{P \times L_1}\right\},$$
$$\mathbb{H}_{\delta,\mathrm{pe}} := \left\{h_{\mathrm{q}}((\mathbf{Y} + \mathbf{F})/L_2) \middle| \mathbf{Y} \in [0, 1)^{P \times L_2}\right\}.$$

Note that for any $\mathbf{X} \in \mathbb{G}_{\delta,\mathrm{pe}}$, all of its columns are different, and for any different $\mathbf{X}, \mathbf{X}' \in \mathbb{G}_{\delta,\mathrm{pe}}$, it holds that the columns of $\mathbf{X}$ are not a permutation of the columns of $\mathbf{X}'$.

Then we can invoke Lemma B.3, which shows the existence of an attention network $g_{\mathrm{c}}$ and a vector $\mathbf{u}$ such that for any $\mathbf{X} \in \mathbb{G}_{\delta,\mathrm{pe}}$, it holds that any entry of $\mathbf{u}^\top g_{\mathrm{c}}(\mathbf{X})$ uniquely determines $\mathbf{X}$. Similarly, there exists $h_{\mathrm{c}}$ and $\mathbf{v}$ which implement contextual mapping for documents. Now we just need the following pooling functions: for the query side, the pooling function outputs $\mathbf{v}$ and $g_{\mathrm{c}}(\mathbf{X})_{:,1}$; for the document side, the pooling function outputs $\mathbf{u}$ and $h_{\mathrm{c}}(\mathbf{Y})_{:,1}$. The similarity matrix is then given by

$$\begin{bmatrix} \mathbf{u}^\top \mathbf{v} & \mathbf{v}^\top h_{\mathrm{c}}(\mathbf{Y})_{:,1} \\ \mathbf{u}^\top g_{\mathrm{c}}(\mathbf{X})_{:,1} & g_{\mathrm{c}}(\mathbf{X})_{:,1}^\top h_{\mathrm{c}}(\mathbf{Y})_{:,1} \end{bmatrix}$$

In particular, the off-diagonal entries of the similarity matrix are enough to determine the query-document pair. Therefore we can further use MLP scorers to approximate the ground-truth scoring function.

## C. Proof of Theorem 3.2

To prove Theorem 3.2, we first construct an empirical dataset on which we show a simple dot-product dual encoder has a large approximation error based on a rank argument. This empirical dataset can then be extended to a distribution on $[0, 1]^{P \times L}$.

Here we let $L_1 = L_2 = L$, i.e., all queries and documents have the same number of tokens. The set of queries is simply $\mathcal{Q} := \{0, 1\}^{P \times L}$, i.e., there are $2^{PL}$ queries, each of them has dimension $P \times L$, and each coordinate of them can be either 0 or 1. The set of documents is also given by $\mathcal{D} := \{0, 1\}^{P \times L}$. Given a query $\mathbf{X} \in \mathcal{Q}$ and a document $\mathbf{Y} \in \mathcal{D}$, define the ground-truth score as

$$K^*(\mathbf{X}, \mathbf{Y}) := \mathrm{tr}(\mathbf{X}^\top \mathbf{Y}) \tag{6}$$

Let $\mathbf{K}^* \in \mathbb{R}^{2^{PL} \times 2^{PL}}$ denote the matrix of ground-truth scores between all query-document pairs. We will show the following result.

**Lemma C.1.** *Let $T_1 : \mathbb{R}^{P \times L} \to \mathbb{R}^O$ denote an arbitrary function that maps a query $\mathbf{X} \in \mathcal{Q}$ to an $O$-dimensional vector, and let $T_2 : \mathbb{R}^{P \times L} \to \mathbb{R}^O$ denote an arbitrary function that maps a document $\mathbf{Y} \in \mathcal{D}$ to an $O$-dimensional vector. Given $\mathbf{X} \in \mathcal{Q}$ and $\mathbf{Y} \in \mathcal{D}$, define the dot-product DE score as $K^{\mathrm{de}}(\mathbf{X}, \mathbf{Y}) = T_1(\mathbf{X})^\top T_2(\mathbf{Y})$, and let $\mathbf{K}^{\mathrm{de}} \in \mathbb{R}^{2^{PL} \times 2^{PL}}$ denote the matrix of DE scores for all query-document pairs. If $O \leq PL - 1$, then the mean square error between $\mathbf{K}^*$ and $\mathbf{K}^{\mathrm{de}}$ is at least $1/16$:*

$$\frac{1}{2^{2PL}} \|\mathbf{K}^* - \mathbf{K}^{\mathrm{de}}\|_F^2 \geq \frac{1}{16}.$$

To prove Lemma C.1, we first show the following linear algebra fact.

**Proposition C.2.** *Let $I_n$ denote the $n$-by-$n$ diagonal matrix, and let $J_n$ denote the $n$-by-$n$ matrix whose entries are all $1$. For $\lambda > 0$, the matrix $\lambda I_n + J_n$ has rank $n$; its top eigenvalue is $\lambda + n$, while the remaining $n - 1$ eigenvalues are $\lambda$.*

*Proof.* First consider the matrix $J_n$. Let $\mathbf{1}_n$ denote the $n$-dimensional vector whose entries are all 1; it is an eigenvector of $J_n$ with eigenvalue $n$. Moreover, $J_n$ also has eigenvalue 0; the corresponding eigenspace is given by $\{\mathbf{z} \in \mathbb{R}^n | \sum_i z_i = 0\}$, which has dimension $n - 1$. As a result, the eigenvalue 0 has multiplicity $n - 1$.

Moreover, note that for any $n$-by-$n$ matrix $\mathbf{A}$ with eigenvalue $\mu$, the matrix $\lambda I_n + \mathbf{A}$ has an eigenvalue $\lambda + \mu$. Consequently, the matrix $\lambda I_n + J_n$ has eigenvalue $\lambda + n$ with multiplicity 1, and eigenvalue $\lambda$ with multiplicity $n - 1$. $\qquad \square$

Next we prove the following properties of $\mathbf{K}^*$ using Proposition C.2.

**Lemma C.3.** *It holds that $\mathbf{K}^*$ has rank $PL$; its top eigenvalue is $2^{PL-2}(PL+1)$, while the remaining $PL - 1$ eigenvalues are $2^{PL-2}$.*

*Proof.* Let $\mathbf{U} \in \mathbb{R}^{2^{PL} \times PL}$ denote the matrix whose rows are obtained by flattening elements of $\{0, 1\}^{P \times L}$ (i.e., the query set $\mathcal{Q}$ and document set $\mathcal{D}$). It then holds that $\mathbf{K}^* = \mathbf{U}\mathbf{U}^\top$. We will analyze the spectrum of $\mathbf{K}^*$ by considering $\mathbf{U}^\top \mathbf{U}$, since it has the same eigenvalues as $\mathbf{U}\mathbf{U}^\top$.

We claim that $\mathbf{U}^\top \mathbf{U} = 2^{PL-2}(I_{PL} + J_{PL})$. First consider diagonal entries of $\mathbf{U}^\top \mathbf{U}$. For any $1 \leq i \leq PL$, it holds that $\mathbf{U}_{:,i}$ has half entries equal to 0, and the other half entries equal to 1. As a result, $(\mathbf{U}^\top \mathbf{U})_{i,i} = 2^{PL-1}$. Next we consider off-diagonal entries of $\mathbf{U}^\top \mathbf{U}$. For any $1 \leq i, j \leq PL$ and $i \neq j$, it holds that $U_{k,i} = U_{k,j} = 1$ for $1/4$ of all positions $k$; therefore $(\mathbf{U}^\top \mathbf{U})_{i,j} = 2^{PL-2}$. This proves our claim.

The claim of Lemma C.3 then follows from Proposition C.2. $\qquad \square$

Now we can prove Lemma C.1

*Proof of Lemma C.1.* Let $T_1 : \mathbb{R}^{P \times L} \to \mathbb{R}^O$ denote an arbitrary mapping; in particular, it could represent a Transformer with positional encodings which maps a query $\mathbf{X} \in \mathcal{Q}$ to an $O$-dimensional embedding vector. Furthermore, let $T_1(\mathcal{Q}) \in \mathbb{R}^{2^{PL} \times O}$ denote the embeddings of all queries given by $T_1$. Similarly, let $T_2 : \mathbb{R}^{P \times L} \to \mathbb{R}^O$ denote an arbitrary mapping which represents the document encoder, and let $T_2(\mathcal{D}) \in \mathbb{R}^{2^{PL} \times O}$ denote embeddings of all documents given by $T_2$. The matrix of dot-product DE scores is then given by $\mathbf{K}^{\mathrm{de}} := T_1(\mathcal{Q})T_2(\mathcal{D})^\top$.

By definition, $\mathbf{K}^{\mathrm{de}}$ has rank at most $O$. If $O \leq PL - 1$, then Lemma C.3 implies that

$$\frac{1}{2^{2PL}} \|\mathbf{K}^* - \mathbf{K}^{\mathrm{de}}\|_F^2 \geq \frac{1}{2^{2PL}} (2^{PL-2})^2 \geq \frac{1}{16}.$$

$\qquad \square$

Then we extend Lemma C.1 to Theorem 3.2.

*Proof of Theorem 3.2.* Recall that the domain of the ground-truth score $K^*$ defined in (6) is $\{0,1\}^{P \times L} \times \{0,1\}^{P \times L}$. We first extend its domain to $[0,1]^{P \times L} \times [0,1]^{P \times L}$ by quantizing the inputs: given $\mathbf{X} \in [0,1]^{P \times L}$, its quantized version $\widehat{\mathbf{X}} \in \{0,1\}^{P \times L}$ is obtained by mapping all entries less than $1/2$ to 0 and other entries to 1. Similarly, given $\mathbf{Y} \in [0,1]^{P \times L}$, we can define its quantized version $\widehat{\mathbf{Y}} \in \{0,1\}^{P \times L}$. We then let $K^*(\mathbf{X}, \mathbf{Y}) = K^*(\widehat{\mathbf{X}}, \widehat{\mathbf{Y}}) = \text{tr}(\widehat{\mathbf{X}}^\top \widehat{\mathbf{Y}})$. Note that $K^*$ defined in this way is not yet continuous; later we will replace it with a continuous ground-truth function, but we will first use $K^*$ below since it simplifies the analysis.

Let $T_1 : \mathbb{R}^{P \times L} \to \mathbb{R}^O$ and $T_2 : \mathbb{R}^{P \times L} \to \mathbb{R}^O$ denote arbitrary mappings. Let

$$\mathbb{M} := \left\{ \mathbf{Z} \in \mathbb{R}^{P \times L} \big| Z_{i,j} = 0 \text{ or } 1/2, 1 \leq i \leq P, 1 \leq j \leq L \right\}.$$

For $\mathbf{Z} \in \mathbb{M}$, let $\mathbb{C}_\mathbf{Z} := \prod_{i=1}^P \prod_{j=1}^L [Z_{i,j}, Z_{i,j} + 1/2]$.

Now we want to find a lower bound on

$$\int_{\mathbf{X} \in [0,1]^{P \times L}, \mathbf{Y} \in [0,1]^{P \times L}} \left( T_1(\mathbf{X})^\top T_2(\mathbf{Y}) - K^*(\mathbf{X}, \mathbf{Y}) \right)^2 d\mathbf{X} d\mathbf{Y}$$

$$= \sum_{\mathbf{Z}, \mathbf{Z}' \in \mathbb{M}} \int_{\mathbf{X} \in \mathbb{C}_\mathbf{Z}, \mathbf{Y} \in \mathbb{C}_{\mathbf{Z}'}} \left( T_1(\mathbf{X})^\top T_2(\mathbf{Y}) - K^*(\mathbf{X}, \mathbf{Y}) \right)^2 d\mathbf{X} d\mathbf{Y}$$

$$= \int_{\mathbf{X} \in \mathbb{C}_\mathbf{0}, \mathbf{Y} \in \mathbb{C}_\mathbf{0}} \sum_{\mathbf{Z}, \mathbf{Z}' \in \mathbb{M}} \left( T_1(\mathbf{X} + \mathbf{Z})^\top T_2(\mathbf{Y} + \mathbf{Z}') - K^*(\mathbf{X} + \mathbf{Z}, \mathbf{Y} + \mathbf{Z}') \right)^2 d\mathbf{X} d\mathbf{Y}, \qquad (7)$$

where we let $\mathbf{0}$ denotes the $P$-by-$L$ matrix whose entries are all 0. Note that in (7), for any $\mathbf{X}, \mathbf{Y} \in \mathbb{C}_\mathbf{0}$, the error can be lower bounded by $2^{2PL}/16$ using the proof of Lemma C.1. Therefore we have

$$(7) \geq \int_{\mathbf{X} \in \mathbb{C}_\mathbf{0}, \mathbf{Y} \in \mathbb{C}_\mathbf{0}} \frac{2^{2PL}}{16} d\mathbf{X} d\mathbf{Y}$$

$$= \frac{2^{2PL}}{16} \cdot \text{vol}(\mathbb{C}_\mathbf{0})^2$$

$$= \frac{1}{16}.$$

As mentioned above, $K^*$ is not continuous, and the final step of the proof is to replace it with a continuous ground-truth function. Previously, we quantize the input by transforming entries less than $1/2$ to 0 and other entries to 1. Now we use the following transformation function $\phi_\tau$: $\phi_\tau(z) = 0$ if $z \leq \frac{1}{2} - \tau$, and $\phi_\tau(z) = 1$ if $z \geq \frac{1}{2} + \tau$, and otherwise $\phi_\tau(z) = \frac{1}{2} + \frac{1}{2\tau}(z - \frac{1}{2})$. Given $\mathbf{X}, \mathbf{Y} \in [0,1]^{P \times L}$, we apply $\phi_\tau$ to every entry of $\mathbf{X}, \mathbf{Y}$ and get $\phi_\tau(\mathbf{X})$ and $\phi_\tau(\mathbf{Y})$, and define $K_\tau^*(\mathbf{X}, \mathbf{Y})$

$$K_\tau^*(\mathbf{X}, \mathbf{Y}) := \text{tr}(\phi_\tau(\mathbf{X})^\top \phi_\tau(\mathbf{Y})).$$

Note that $K_\tau^*$ is continuous for any $\tau$, and as $\tau$ goes to 0, it holds that $K_\tau^*$ becomes arbitrarily close to $K^*$ in $\ell_2$ distance. Therefore there exists a small enough $\tau$ such that $K_\tau^*$ satisfies the requirements of Theorem 3.2. $\square$

