# OpenReview forum: "Efficient Document Ranking with Learnable Late Interactions"
_ICML.cc/2024/Workshop/WANT — WANT@ICML 2024 Poster_

### Official Review · Reviewer_ZTMq · 2024-06-13
**Summary of LITE paper**

**Confidence:** 5

**Summary:**

Paper is about document rankings. Authors propose novel LITE (Learnable Late Interaction) model that has several advantages(generalization, latency, storage) than previous existing methods. In general, proposed LITE models offer a novel approach that combines the benefits of CE and DE models, providing high-quality relevance predictions with reduced latency and storage requirements.

**Strengths:**

•  There is a clear summary of the research objectives, methods and novelty in abstract section.
•  In the introduction section all necessary information is provided, scientific/research question is well-defined and contribution is also explained.
•  Background section summarizes relevant previous studies and findings, identifies gaps in previous studies
•  LITE Scorers section clarifies everything that's needed
•  There are used several datasets and methods(more than one) in the Experiments section to compare previous methods to LITE and prove that it has better performance.
•  Conclusion sums up everything very well

**Weaknesses:**

1) While the paper shows promising zero-shot results, the generalization capabilities of LITE across highly diverse or unseen domains are still uncertain.
2) The performance of the LITE model might heavily rely on the quality of pre-trained embeddings. The paper does not extensively explore the impact of different pre-trained models (e.g., BERT, RoBERTa, GPT) on the final performance of LITE.

---

### Meta-Review · Area_Chair_Kher · 2024-06-17

**Recommendation:** Accept (Poster)
**Confidence:** 3

**Metareview:**

This manuscript targets the application of predicting query-document relevance and introduces a learnable yet efficient strategy to automatically trade off the latency-quality. The paper is written in good shape: it has sufficient technical novelty and extensive numerical results to fit the interest of the workshop. Authors are encouraged to conduct more ablation studies to justify the effectiveness of the proposed LITE, e.g., in terms of domain generalization.

---

### Decision · Program_Chairs · 2024-06-17

**Decision:**

Accept (Poster)

**Comment:**

We thank the authors for their time and contribution to WANT and we are pleased to share that after the reviewing process the paper has been accepted. Congratulations! We encourage the authors to consider reviewers' feedback for the improvement of the camera-ready version. We hope to see you in person at the workshop and brainstorm on efficient training research together!